# CBCT Evaluation of Sticky Bone in Horizontal Ridge Augmentation with and without Collagen Membrane—A Randomized Parallel Arm Clinical Trial

**DOI:** 10.3390/jfb13040194

**Published:** 2022-10-19

**Authors:** Jane Belinda Tony, Harinath Parthasarathy, Anupama Tadepalli, Deepa Ponnaiyan, Ahmed Alamoudi, Mona Awad Kamil, Khalid J. Alzahrani, Khalaf F. Alsharif, Ibrahim F. Halawani, Mrim M. Alnfiai, Lakshmi Ramachandran, Thodur Madapusi Balaji, Shankargouda Patil

**Affiliations:** 1Department of Periodontics, SRM Dental College, Ramapuram, Chennai 600089, India; 2Oral Biology Department, Faculty of Dentistry, King Abdulaziz University, Jeddah 22254, Saudi Arabia; 3Department of Preventive Dental Science, College of Dentistry, Jazan University, Jazan 45142, Saudi Arabia; 4Department of Clinical Laboratories Sciences, College of Applied Medical Sciences, Taif University, P.O. Box 11099, Taif 21944, Saudi Arabia; 5Department of Information Technology, College of Computers and Information Technology, Taif University, P.O. Box 11099, Taif 21944, Saudi Arabia; 6Tagore Medical and Dental College, Rathinamangalam, Vandalur, Chennai 600127, India; 7College of Dental Medicine, Roseman University of Health Sciences, South Jordan, UT 84095, USA

**Keywords:** collagen membrane, guided bone regeneration horizontal ridge augmentation, injectable-platelet rich fibrin (i-PRF), sticky bone

## Abstract

Guided bone regeneration (GBR) is a reliable technique used to treat ridge deficiencies prior or during implant placement. Injectable-platelet rich fibrin (i-PRF) laced with a bone substitute (sticky bone) has heralded the way for advancing the outcomes of bone regeneration. This study evaluated the efficacy of sticky bone in horizontal ridge augmentation with and without collagen membrane. A total of 20 partially edentulous patients (Group-I n = 10; Group-II n = 10) that indicated GBR were included, and the surgical procedure was carried out. In Group-I, the sticky bone and collagen membrane were placed in ridge-deficient sites and Group-II received only sticky bone. At the end of 6 months, 20 patients (Group-I (n = 10); Group-II (n = 10)) completed the follow-up period. A CBCT examination was performed to assess changes in the horizontal ridge width (HRW) and vertical bone height (VBH). A statistically significant increase in HRW (*p <* 0.05) was observed in both groups with mean gains of 1.35 mm, 1.55 mm, and 1.93 mm at three levels (crest, 3 mm, and 6 mm) in Group-I and 2.7 mm, 2.8 mm, and 2.6 mm at three levels in Group-II. The intergroup comparison revealed statistical significance (*p <* 0.05) with respect to HRW and KTW (Keratinised tissue width) gains of 0.775 at the 6-month follow-up. Sticky-bone (Xenogenic-bone graft + i-PRF) served as a promising biomaterial in achieving better horizontal bone width gain.

## 1. Introduction

Physiologic Alveolar ridge resorption following tooth loss leads to a range of ridge deficiencies. The majority of the dimensional alterations occur during the first three months and can persist up to 5 years [1,2], with the buccal or facial bone bearing the major brunt of the resorption process. Ridge augmentation procedures are often required for optimal implant placement, which aids in the long-term preservation of the peri-implant hard and soft tissue. Guided bone regeneration (GBR) procedures frequently utilize barrier membranes and bone substitutes and are well-documented and widely used for augmenting deficient alveolar ridges.

Bone substitutes are becoming more widely accepted as alternatives to autogenous bones for ridge augmentation procedures, minimising the morbidity associated with autogenous graft harvesting [3,4,5,6]. Most grafting materials serve as a scaffold and aid in osteoconduction by allowing the proliferation of the surrounding osteogenic cells, maintaining space and stabilising the blood clot. Anorganic bovine bone mineral (ABBM) has been frequently used in Horizontal Ridge augmentation and is the most well-documented bone substitute [7,8]. ABBM possesses excellent osteoconductive properties and the propensity to facilitate neovascularisation [9]. Accordingly, there is sufficient evidence in the literature explaining the advantages of using ABBM in ridge augmentation procedures. However, due to the absence of biological components such as growth factors and signalling molecules, ABBM lacks the competence to induce osteoinduction and osteogenesis and, thus, fails to synchronise with the osteogenic rate, resulting in a prolonged graft-healing period [10]. Hence, integrating bone graft materials with biomimetics such as growth factors may minimise healing time, aid in osteoinductive bone remodelling, and improve volumetric augmentation and the quality of the new bone formed. [11] Exogenous growth factors such as Recombinant human platelet-derived growth factor-BB (rhPDGF-BB), Bone morphogenetic protein (BMP-2) and (BMP-7), and Growth differentiation factor-5 (GDF-5) have been proven to enhance bone regeneration [12].

Second generation Autologous platelet concentrates such as Platelet-rich fibrin (PRF) and Concentrated growth factors (CGF) were proposed to be used as an autologous fibrin glue to create a bone graft matrix loaded with growth factors, known as "sticky bone" [13]. 

In 2015, Mourao et al., stated that platelet-rich fibrin can be replaced by Injectable platelet-rich (i-PRF) fibrin, which has the ability to release a variety of growth factors with increased expression of Platelet-derived growth factor (PDGF), Transforming growth factor (TGF), Collagen 1, and Fibroblast migration for accelerating bone regeneration [14,15]. This injectable form is combined with bone grafts and collagen matrices and can be directly injected into periodontal pockets to improve the clinical outcomes of various regenerative procedures.

The sticky bone, according to Sohn et al., 2015 [16], may function as a framework or protective membrane, both externally and internally, to the periosteum and the alveolar bone. Sticky bone has excellent handling properties, is remarkably mouldable, and has good adherence to the defect site. Further, it minimises the micro and macro motility of the grafted bone, enabling favourable bone augmentation throughout the healing period without the use of a GBR membranes or titanium meshes.

Various GBR membranes such as Type-1 Collagen, poly-lactic acid, polyglycolic acid, polyurethane, and dura mater are among the resorbable membranes available, with collagen membrane being the most popular and widely used with extensive documentation.

The literature shows that collagen membrane exhibits 22 percent to 32 percent early membrane exposure, [17,18,19] which tends to lose its integrity in a week after exposure to the proteolytic environment of the oral cavity, and is easily infected, leaving the graft material vulnerable and eventually leading to graft loss.

Sticky bone bypasses the need for a membrane, and the fibrin network drives platelets and leukocytes to release signalling molecules and drastically increase the regeneration of hard and soft tissue. The fibrin interconnectivity of the sticky bone also diminishes soft tissue and epithelial cell ingrowth [16]. Cordaro et al., reported that harnessing a barrier membrane increases the risk of bone graft exposure, dramatically reducing the treatment outcome [20]. The pitfalls of using a collagen membrane include a high cost, membrane exposure, the need for surgical competence to fix the membrane with membrane tags and its associated additional expenses, and the overall increase in surgical time. Hence, we hypothesise that using a collagen membrane along with ABBM+i-PRF (Sticky Bone) will not provide additional clinical benefits. Therefore, the current study sought to assess the effectiveness of using sticky bone with and without a collagen membrane in augmenting horizontal posterior mandibular ridge defects.

## 2. Materials and Methods

This randomised parallel arm clinical trial was conducted with approval of the institutional scientific and Ethical committee (SRMDC/IRB/2019/MDS/No.505A) before the commencement of the study and was registered with clinicaltrials.gov.in (CTRI/2020/06/025567). Sample size calculation was performed based on results obtained from a study by Mohammed Atef. et al in February 2020 [21]. The test was performed using UCn Software (SPSS for Windows, version 17, Chicago, IL, USA) with 80% power, alpha error of 5%, Sample Mean of 3.9, and Standard Deviation of 0.9. Based on the sample size determination, a total of 20 sites were recruited for the study in which 10 sites requiring ridge augmentation were operated on with Sticky Bone and covered with Type I collagen membrane (GROUP-I) and 10 sites requiring ridge augmentation were operated on with Sticky Bone alone (GROUP-II), and the statistical significance is considered to be at the 0.05 level. The study population was recruited from the Periodontics and Oral Implantology department, SRM Dental College, Ramapuram campus from January 2020 until September 2020. **Inclusion criteria**: systemically healthy individuals within an age range of 21–60 years presenting with edentulous sites requiring a ridge width gain of 4–5 mm buccolingually and with a residual ridge width of 2–3 mm. The edentulous sites were required to be Kennedy’s Class III, with a post extraction healing of minimum of 6 months. **Exclusion criteria:** patients with debilitating systemic diseases or diseases that had a clinically significant effect on wound healing, had a history of intravenous bisphosphonate/oral bisphosphonate use for > 3 years or history of head and neck radiation therapy/chemotherapy, smokers, pregnant women or lactating mothers, and people possessing allergies to any of the materials or medications used in the current study were excluded.

To establish the keratinised tissue width (KTW) and vestibular depth (VD), custom-made composite resin stents were prepared for each patient with a vertical groove created mid-buccally over the stent to guide consistent reorientation of the periodontal probe (UNC-15).

### 2.1. Radiological Assesment

For Groups I and II, a Cone Beam Computed Tomography (CBCT) scan was performed prior to surgery and 6 months post-surgery. Horizontal Ridge Widths (HRW) were measured at the crest, 3 mm, and 6 mm levels perpendicular to the longitudinal axis of the alveolar crest via tomographic sections. The Vertical Bone Height (VBH) of the ridge was measured from the crest to the mandibular canal.

### 2.2. Randomization

The Patients were randomly allocated using the lottery method. The investigator who evaluated the CBCT measurements was blinded in this study. A total of 20 ridge-deficient sites requiring ridge augmentation were randomized into 2 groups of 10 sites each.

### 2.3. Surgical Procedure

All surgical procedures were carried out by a single, trained periodontal surgeon (H.P). For both Group I and Group II, local anaesthesia was administered using 2% lignocaine with adrenaline. After performing a mid-crestal incision through the keratinized gingiva, a full-thickness flap was raised on the buccal and lingual side. To enable osteoprogenitor cell migration, multiple neo-angiogenetic cortical holes were made in the recipient bone bed. A total of 10 mL of venous blood obtained from the antecubital vein in non-coated plastic vacutainers was immediately spun in a centrifuge at a pre-programmed spin to obtain i-PRF (700 rpm for 3 min with 60 g force) (Dentifuge LC-100). The prepared i-PRF was mixed with ABBM (Bio-OSS^TM^, Geistlich®, Zurich, Switzerland) (Figure 1); polymerisation takes approximately 5–10 min, resulting in sticky bone formation [22,23,24]. In Group I, sticky bone was adapted and moulded over the defect to the level of the adjacent neighbouring buccal bony contour. A bi-layered cross-linked collagen membrane (Cologide™, Cologenesis Healthcare Pvt Limited) of adequate size was chosen and cut to fit the defect size, and it was employed to cover the bone grafts with a 2 mm overlap buccally (Figure 2A–H) and on the lingual side the membrane was tucked inside the lingual flap; in Group-II, the same protocol as Group-I was followed, with the placement of the prepared sticky bone without the collagen membrane (Figure 3A–H). Tension-free primary closure was achieved with simple, interrupted (3-0 Vicryl, Ethicon) sutures in both groups. Six months after the augmentation, cone beam-computed tomographic (CBCT) images were taken.

### 2.4. Postoperative Wound Care and Medication

Patients were administered with antibiotics (Amoxicillin 250 mg, three times daily) and analgesics (Ibuprofen+paracetamol) for up to five days after surgery. For the first four weeks after surgery, patients were told not to brush the surgical site and chlorhexidine mouth wash (0.12%) was prescribed as an adjunct to oral hygiene maintenance. All patients experienced uneventful wound healing. There was no wound dehiscence during the healing process. Patients had a stringent postoperative supportive care regimen and underwent reviews beginning one week after surgery, continuing every two weeks until eight weeks had passed post-surgery. Patients were then recalled once a month until six months as passed for assessment.

### 2.5. Statistical Analysis

IBM SPSS was used to carry out the statistical analysis with 25.0 statistics programme. Using an independent samples T-test, the groups’ distributions of age and gender were compared. In order to compare and evaluate the relationship between the categories, the Chi-Square Test was employed. An unpaired t-test was developed to compare the values between the two groups; Values within the groups were compared using the paired sample t-test. The level of statistical significance was fixed at *p <* 0.05.

## 3. Results

The primary outcome of our study was the assessment of the inter and intra group changes in the horizontal ridge width using Cone Beam Computed Tomography (CBCT). The secondary outcomes analysed were the changes in vertical bone height, changes in vestibular depth, and changes in the keratinized tissue width. A total of 20 patients (11 males and 9 females) with a mean age of 36.1 years participated in the current investigation (Table 1).

The intragroup comparisons (Group I- Sticky Bone and Collagen Membrane) of the primary clinical parameters such as the HRW at various levels (at the crest, at 3 mm, and at 6 mm) showed a statistically significant difference at 6 months of healing compared to the baseline with the highest bone gain of 1.93 mm at the 6 mm level. There was a statistically significant improvement in the VD and KTW at 6 months compared to the baseline. 

The intragroup comparisons (Group II- Sticky Bone) of the primary clinical parameters such as the HRW at various levels (at the crest, at 3 mm, the at 6 mm) showed a statistically significant difference at 6 months of healing compared to the baseline, with the highest bone gain of 2.8 mm at the 6 mm level. There was a statistically significant improvement in the VBH and KTW at 6 months compared to the baseline (Table 2).

The intergroup comparisons between Group-I and Group-II for various clinical parameters such as the Horizontal ridge width (HRW), Vertical bone height (VBH), Vestibular depth (VD), and width of Keratinised tissue (KTW) at baseline showed no statistical significance between Group-I and Group-II. However, the results at 6 months showed a statistically significant improvement in Group-II. Even the KTW gain was statistically significant between the groups. (Table 3)

## 4. Discussion

Changes in the alveolar ridge dimensions post extraction have previously been reported [25,26]. According to Tan et al., in (2012) [27], vertical hard tissue resorption ranges from 11 to 22 percent, whilst horizontal bone loss ranges from 29 to 63 percent, with two-thirds of it lost in the first three months post-tooth extraction [27,28]. A lack of a sufficient quantity and quality of residual alveolar ridges can seriously undermine optimum primary stability and osseointegration. Three-dimensional implant positioning, an adequate bone quantity, a keratinised gingiva volume to promote adequate biological width establishment, and the establishment of a soft tissue seal around the implant–crown interface are all the critical determinants for successful implant osseointegration and long-term function [29].

The current study aimed at evaluating the outcomes of the use of sticky bone with and without a collagen membrane, following horizontal alveolar ridge augmentation, which was determined by using CBCT. The membrane employed in this investigation was a bilayer-cross-linked collagen membrane. 

In the intra group (Group-I: sticky bone + collagen membrane) comparison of various clinical parameters—such as the HRW at various levels (at the crest, at 3 mm, and at 6 mm), the VBH, the KTW in Group-I showed a statistically significant improvement in all the clinical parameters including the HRW gain, VD, and KTW. The results are comparable with those of Aboelela et al. (2021) [30] who evaluated the efficacy of anorganic bovine bone and autogenous bone combination with CGF and a native collagen membrane and showed a mean gain of 2.4 mm, whereas, in our study, the mean gain achieved was 1.93 mm. In a study by Geurs et al. (2008) [31], bovine bone chips and a biological carrier with a synthetic PGA membrane exhibited a mean gain of 2.9 mm from the baseline. Further, Eskan et al. in (2017) [32] showed a mean gain of 2.6 mm in lateral ridge augmentation using an allogenic Cancellous (CAN) graft particle and an alloplastic resorbable barrier membrane. The variation in the biomaterials utilised in the above-mentioned studies and our study could explain the noticeable increase in the horizontal mean ridge width gain. However, using particulate graft materials and a membrane, all of the previous studies, including ours, demonstrated a considerable increase in HRW gain. The science behind new bone formation lies in the fact that the grafted bone material acts as a scaffold that favours and promotes the neovascularisation and the ingrowth of osteogenic cells over and into the matrix; further, the ABBM (Bio-OSS^TM^, Geistlich®, Switzerland) used in this study was combined with i-PRF and used in the form of sticky bone, which could have led to the slow release of critical growth factors such as PDGF, VEGF (vascular endothelial growth factor), IGF (insulin like growth factor), and FGF (fibroblast growth factor), which have shown to favour new bone formation. With the additional use of a collagen barrier membrane, epithelial exclusion and space protection is expected, which in turn favours optimal treatment outcomes. 

The intragroup comparisons (Group II- Sticky Bone) of various clinical parameters such as the HRW at various levels, the VBH, and the KTW showed a statistically significant improvement with a mean horizontal gain of 2.8 mm and a VBH gain of 1 mm.

The results of the present study cannot be compared directly with any other study as this the first study of its kind using sticky bone alone for horizontal ridge augmentation. However, Staedt et al. (2020), using an animal model, histologically demonstrated significant new bone formations that were further reinforced using enzymatic assays such as Bone acid phosphatase (BAcPH) and total alkaline phosphatase (TAlPh) [33]. Most of the studies using particulate xenogenic grafts for horizontal ridge augmentation have utilised a GBR membrane to improve clinical outcomes; however, we have proposed that optimal bone gain can be achieved without the use of a membrane.

The Bio-OSS^TM^ (Geistlich®, Switzerland) bone graft has been a very successful biomaterial used in the augmentation of resorbed ridges. It acts as a scaffold and favours new bone formation by osteoconduction; further, the addition of i-PRF to the sticky bone matrix may have led to the adherence of the grafted material to the recipient sites without micro and macro movements, and the PRF matrix prevents early epithelial ingress onto the defect site leading to significant new gains in both horizontal and vertical dimensions.

The intergroup comparisons showed an additional mean HRW gain of 0.9 mm in Group-II (Sticky bone alone), which was statistically significant. Further, there was a statistically significant improvement in the KTW (again, in the solely sticky bone group). To the best of our knowledge, this is a novel approach, wherein we have analysed and compared the efficacy of sticky bone with and without a collagen membrane. The results of the present study could not be directly compared with previous studies in the literature. However, Lee et al. (2013), in his study comparing a tooth bone graft with and without a membrane, determined a similar bone gain in both the groups and further carefully elucidated that the use of an additional membrane did not benefit the clinical outcomes [34]. Further, Staedt et al. (2020), in a surgically created defect in an animal model, compared Bio-OSS^TM^ (Geistlich®, Switzerland) with and without a collagen membrane and concluded that though a bone graft with a membrane showed significantly earlier bone remodelling, there was no histological difference in the new bone formation between the groups [33]. In the present study, the additional gain in the horizontal ridge with the sticky bone alone (group-II ) could be attributed to the collapsing nature of the GBR membrane causing a lack of space maintenance and more aggressive remodelling in group-I (sticky bone + collagen membrane)

In the introduction, we proposed that there would not be an additional benefit of using membranes, which has been proven from the results of our study. According to numerous studies, a GBR membrane is intended to protect and encapsulate the graft material during the sensitive bone-remodelling phase and its integration with the native bone [35]. However, there are several drawbacks to using GBR membranes, such as the difficulty in membrane stabilisation, its exorbitant cost, and rapid and unpredictable disintegration [36] which can result in a weakened barrier effect, as well as the presence of chemical residues, which might elicit an undesirable host immuno-inflammatory response during the healing phase. Membrane exposure compounds the problem even further by causing infections, which impacts the therapeutic outcome. In a systematic review and meta-analysis by Garcia et al. (2018), it was stated that the use of a collagen membrane for GBR sites with membrane exposure led to significantly poorer horizontal bone gain [37]. Further, Lee et al. in 2013 and Eskan et al. in 2017 also concluded that early membrane exposure led to a significant reduction in the alveolar ridge width. We further state that in our study no membrane exposure was noticed, and all the sites healed uneventfully [32,34].

The encouraging results obtained in both groups are due to the biological properties of i-PRF, which acts as biological glue by keeping the graft particles intact and at the same time allowing for neovascularisation during healing. Furthermore, the various growth factors released from the PRF such as PDGF, EGF (epidermal growth factor), IGF, FGF, and VEGF help in promoting cellular proliferation, thereby increasing vascularity at the surgical site. The absence of a constrictive effect of the collagen membrane, the easy re-approximation of the flap, and the unjeopardized vascularity in group-II resulted in comparatively better outcomes. The CBCT analysis resulted in a reliable, non-invasive assessment of the changes in bone width and height.

The study limitations include a small sample size, a longer follow-up, and a lack of histological evaluation of the new bone formed.

## 5. Conclusions

Therefore, within the limits of the current study, we conclude that both the groups showed significant improvement in all the clinical parameters assessed, and we propose that a combination of ABBM+i-PRF (Sticky bone) can be predictably used without the need for a collagen barrier membrane as it did not provide any additional benefits regarding horizontal ridge augmentation in the current study.

## Figures and Tables

**Figure 1 jfb-13-00194-f001:**
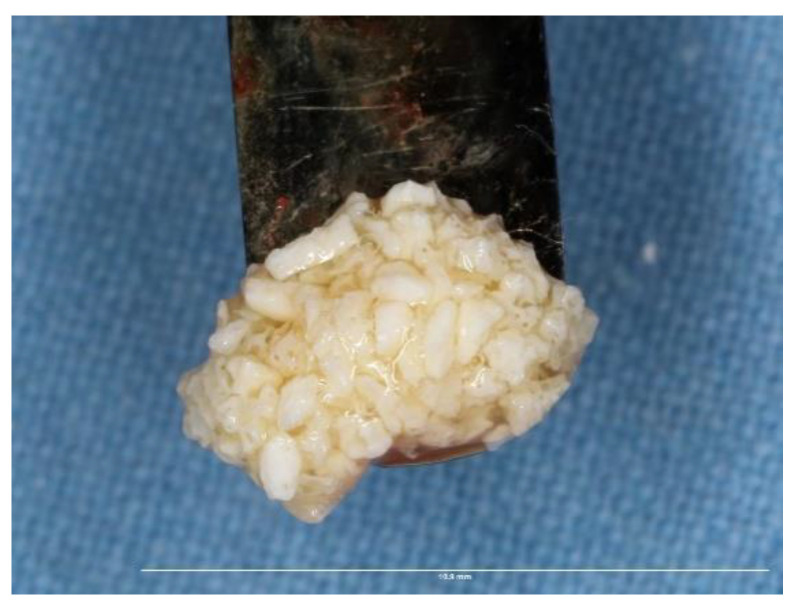
Prepared sticky bone.

**Figure 2 jfb-13-00194-f002:**
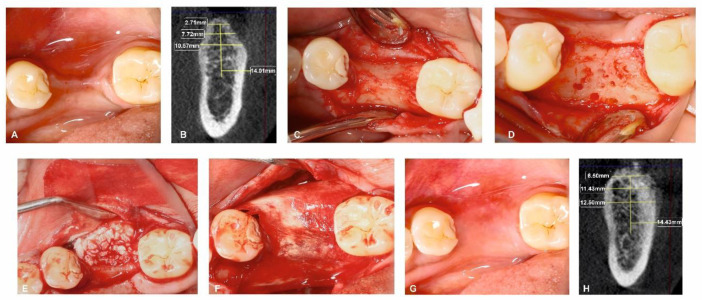
Surgical management of horizontal ridge defect (group-I). (**A**) Preoperative horizontal ridge deficient site; (**B**) HRW and VBH measurements assessed at crest, at 3 mm, and at 6 mm in baseline; (**C**) Flap elevation; (**D**) Decortications performed in the surgical site; (**E**) Sticky bone placement; (**F**) Collagen membrane adapted over the sticky bone; (**G**) 6-month follow-up; (**H**) HRW and VBH measurement assessed at crest, at 3 mm, and at 6 mm at 6 months.

**Figure 3 jfb-13-00194-f003:**
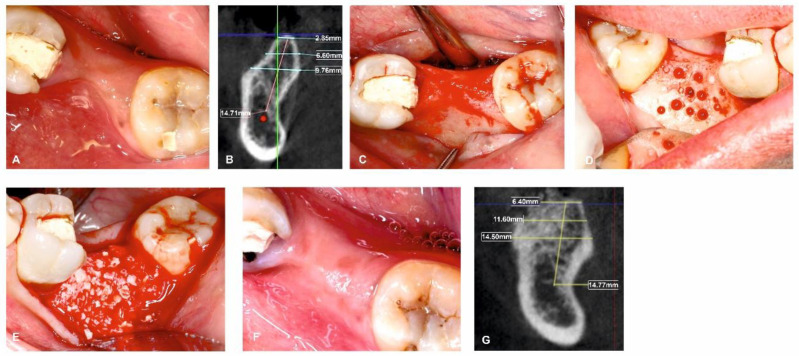
Surgical management of horizontal ridge defect (group-II). (**A**) Preoperative horizontal ridge deficient site; (**B**) HRW and VBH measurements assessed at crest, at 3 mm, and at 6 mm in baseline; (**C**) Flap elevation; (**D**) Decortications performed in the surgical site; (**E**) Sticky bone placement; (**F**) 6-month follow-up; (**G**) HRW and VBH measurement assessed at crest, at 3 mm, and at 6 mm at 6 months.

**Table 1 jfb-13-00194-t001:** Demographic characteristics of the included patients.

			Group	
			Sticky Bone + Membrane (Group I)	Sticky Bone (Group II)
Gender	Male	Count	3	8
		% Within Group	30.0%	80.0%
	Female	Count	7	2
		% Within Group	70.0%	20.0%
Age (Mean ± SD)				
			34.5 ± 8.66	37.7 ± 4.47

**Table 2 jfb-13-00194-t002:** Intra group comparison of the parameters at baseline and 6 months among Group-I and Group-II.

Clinical Parameters	“Group-I” (Sticky Bone+ Collagen Membrane)			*p* Value	“Group-II” (Sticky Bone)			*p* Value
Parameters	“Baseline”	“6 Months”	“Baseline- 6 Months”	“Baseline- 6 Months”	“Baseline”	“6 Months	“Baseline- 6 Months”	“Baseline- 6 Months”
HRW AT CREST	2.90 ± 0.316	4.25 ± 1.165	−1.375	**0.020** ^a^	3.20 ± 0.789	5.90 ± 0.994	−2.7	<**0.001** ^a^
HRW AT 3 mm	4.20 ± 1.033	5.75 ± 1.581	−1.375	**0.001** ^a^	6.10 ± 1.595	8.90 ± 1.853	−2.8	<**0.001** ^a^
HRW AT 6 mm	6.70 ± 1.947	8.63 ± 2.134	−2	**0.001** ^a^	8.30 ± 1.767	10.90 ± 1.524	−2.6	<**0.001** ^a^
VBH	13.20 ± 1.317	13.88 ± 1.727	−0.875	0.111	12.50 ± 2.461	13.50 ± 2.677	−1	<**0.001** ^a^
VD	10.60 ± 0.699	10.88 ± 0.835	−0.5	**0.033** ^a^	10.90 ± 0.876	11.20 ± 0.919	−0.3	0.081
KTW	4.10 ± 0.876	4.88 ± 0.835	−0.625	**0.049** ^a^	3.60 ± 0.516	4.10 ± 0.568	−0.5	**0.015** ^a^

^a^*p*-value ≤ 0.05 considered statistically significant”; HRW: horizontal ridge width; VBH: vertical bone height; KTW: keratinised tissue width; VD: vestibular depth.

**Table 3 jfb-13-00194-t003:** Inter-Group Comparison Of The Clinical And Radiographic Parameters At Baseline And 6 Months.

	Clinical Parameter	Time Points	Mean Difference	*p*-Value
Group-I Vs Group-II	HRW At Crest	Baseline	−0.30	0.279
		6 Months	−1.65	**0.005** ^a^
	HRW At 3 mm	Baseline	−1.90	**0.005** ^a^
		6 Months	−3.15	**0.002** ^a^
	HRW At 6 mm	Baseline	−1.60	0.070
		6 Months	−2.27	**0.018** ^a^
	VBH	Baseline	0.883	0.438
		6 Months	1.09	0.737
	VD	Baseline	−0.30	0.408
		6 Months	−0.325	0.449
	KTW	Baseline	0.50	0.137
		6 Months	0.775	**0.032** ^a^

^a^*p*-value ≤ 0.05 considered statistically significant”; HRW: horizontal ridge width; VBH: vertical bone height; KTW: keratinised tissue width; VD: vestibular depth.

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
