# Peer review of "CBCT Evaluation of Sticky Bone in Horizontal Ridge Augmentation with and without Collagen Membrane—A Randomized Parallel Arm Clinical Trial"

_jfb, 2022, doi:10.3390/jfb13040194_

Round 1

Reviewer 1 Report

I appreciate the topic of the article as being a very up-to-date one and also of great interest due to the clinical aspects followed in the study. Nevertheless, I have some observations to highlight, as follows:

-the material contains writing errors that must be corrected

-regarding the study design, it is a very important issue that must be taken into account: when analyzing potential GBR techniques, it is compulsory, before recommending one material/technique or another, to have a very accurate and complete analysis of the results, of factors influencing the outcome and of caliber of the study. In this particular case, a volumetric measurement has nothing to do with assessing the quality of the regenerated bone, especially when analyzing xenograft exclusively, which has a slow resorption/remodeling rate.

-supporting the previous observation, I strongly recommend to change the conclusion of the study, as one cannot conclude the effectiveness of a GBR technique only thought out CBCT measurements, excluding the gold standard biopsy assessment. The authors conclude that: “it can be consummated that a combination of ABBM+I-PRF (Sticky bone) can be predictably used without the need of a membrane as it did not provide any additional benefit for Horizontal Ridge augmentation procedures”. I appreciate this conclusion as being inappropriate, as it is not being scientifically sustained, a volumetric analyze at 6 months after horizontal ridge augmentation in mandible is not fully relevant.

-the quality of the pictures is very low, measurements done on CBCT slices are not readable

-reference section also contains formatting errors and it is not up-to-date (a lot of old articles)

I consider the title of the study and also the conclusions as being more comprehensive than the study itself. Therefore, considering the limitations of the study design, I would recommend changing these referring exclusively to the volumetric assessment using CBCT.

Author Response

Reviewer 1:

I appreciate the topic of the article as being a very up-to-date one and also of great interest due to the clinical aspects followed in the study. Nevertheless, I have some observations to highlight, as follows:

-the material contains writing errors that must be corrected

Reply:  The writing errors in the material are corrected.

-regarding the study design, it is a very important issue that must be taken into account: when analyzing potential GBR techniques, it is compulsory, before recommending one material/technique or another, to have a very accurate and complete analysis of the results, of factors influencing the outcome and of caliber of the study. In this particular case, a volumetric measurement has nothing to do with assessing the quality of the regenerated bone, especially when analyzing xenograft exclusively, which has a slow resorption/remodeling rate.

Reply: we would like to bring to the kind notice of the reviewer that ,we have not commented anywhere in the manuscript regarding the qualitative analysis of the new bone formed ,as we accept the fact mentioned by the reviewer that quantitative assessment only can be done using CBCT.

-supporting the previous observation, I strongly recommend to change the conclusion of the study, as one cannot conclude the effectiveness of a GBR technique only thought out CBCT measurements, excluding the gold standard biopsy assessment. The authors conclude that: “it can be consummated that a combination of ABBM+I-PRF (Sticky bone) can be predictably used without the need of a membrane as it did not provide any additional benefit for Horizontal Ridge augmentation procedures”. I appreciate this conclusion as being inappropriate, as it is not being scientifically sustained, a volumetric analyze at 6 months after horizontal ridge augmentation in mandible is not fully relevant.

Reply: taking guidance from the reviewer’s comment we are adding CBCT evaluation to the title of the study and has been changed accordingly in the manuscript. Further, this study being part of the dissertation work by PG student and was done during the peak COVID pandemic period , few of the patients did not complete more than 6 months follow-up. Further many studies have shown predictable long term implant success in sites augmented with ABBM and implants placed within 3-6 months of augmentation has the middle and the apical native bone provide adequate stability and predictable osseointegration whilst the coronal augmented bone is still in the stage of active remodeling.

References:

1.According to Buser et al.,2013, he concluded that early implant placement with simultaneous contour augmentation offers high predictability for successful outcomes and good long-term stability of the established facial bone wall. The periapical radiographs yielded stable peri-implant bone levels, with a mean DIB(distance from the implant shoulder to the first bone-to-implant contact) of 0.44 mm.

  1. Accoridng to Nissan et al, 2009, concluded that early Implant placement in the posterior atrophic mandible following augmentation showed a implant survival rate of 95.3% .
  2. According to donos et al.,2008, early implant placement implants placed with simultaneous GBR or a staged procedure, depicted an implant survival rate at the augmented sites from 91.7% to 93.2%.

-the quality of the pictures is very low, measurements done on CBCT slices are not readable

Reply: Have improved the quality of the images and have made the CBCT slices readable.

-reference section also contains formatting errors and it is not up-to-date (a lot of old articles)

Reply: Have formatted the references style and have updated with recent year references.

I consider the title of the study and also the conclusions as being more comprehensive than the study itself. Therefore, considering the limitations of the study design, I would recommend changing these referring exclusively to the volumetric assessment using CBCT.

Reply: Have altered the Title of the study.

Reviewer 2:

This manuscript presented a study about the efficacy of sticky bone in horizontal ridge augmentation with and without membrane. The work has some potential. However, the authors must improve the discussions about the obtained results, according to the comments below.

Introduction: please be clearer about the novelty of this work.

Figure 2: if possible add a scale bar in Figure 2.

Reply: Uploaded the image with scale bar added to it.

Lines 282-287: the authors comment the results and the improvements observed, but not provide an explanation to what can be the probably reason to the observed result. I suggest better discuss this result.  

Reply: Have justified the probable reason for the observed result.

Lines 300-308: also again the authors compared the results with other results from the literature, but not provide a probably reason to the observed result.

Reply: Have justified the probable reason for the observed result.

Reviewer 2 Report

This manuscript presented a study about the efficacy of sticky bone in horizontal ridge augmentation with and without membrane. The work has some potential. However, the authors must improve the discussions about the obtained results, according to the comments below.

Introduction: please be clearer about the novelty of this work.

Figure 2: if possible add a scale bar in Figure 2.

Lines 282-287: the authors comment the results and the improvements observed, but not provide an explanation to what can be the probably reason to the observed result. I suggest better discuss this result.  

Lines 300-308: also again the authors compared the results with other results from the literature, but not provide a probably reason to the observed result.

Author Response

(The authors gave the same response as above.)

Round 2

Reviewer 1 Report

The previous observation is still unsolved: the quality of the pictures is very low, measurements done on CBCT slices are not readable. The authors only modified the contrast, they did not improve the quality of the figures, the numbers are still not readable.

Author Response

Reviewer 1:

The previous observation is still unsolved: the quality of the pictures is very low, measurements done on CBCT slices are not readable. The authors only modified the contrast, they did not improve the quality of the figures, the numbers are still not readable.

Reply: Have improved the quality of the image, with the readable CBCT measurements.

Reviewer 2:

After corrections the manuscript reads well. I suggest publication. 

Reply: We thank the reviewer for the positive comment

Reviewer 2 Report

After corrections the manuscript reads well. I suggest publication. 

Author Response

(The authors gave the same response as above.)
